# Predictors of Mental Health Literacy among Parents, Guardians, and Teachers of Adolescents in West Malaysia

**DOI:** 10.3390/ijerph20010825

**Published:** 2023-01-01

**Authors:** Picholas Kian Ann Phoa, Asrenee Ab Razak, Hue San Kuay, Anis Kausar Ghazali, Azriani Ab Rahman, Maruzairi Husain, Raishan Shafini Bakar, Firdaus Abdul Gani

**Affiliations:** 1Department of Psychiatry, School of Medical Sciences, Universiti Sains Malaysia Health Campus, Kota Bharu 16150, Kelantan, Malaysia; 2Biostatistics and Research Methodology Unit, School of Medical Sciences, Universiti Sains Malaysia Health Campus, Kota Bharu 16150, Kelantan, Malaysia; 3Department of Community Medicine, School of Medical Sciences, Universiti Sains Malaysia Health Campus, Kota Bharu 16150, Kelantan, Malaysia; 4Department of Psychiatry and Mental Health, Sultan Haji Ahmad Shah Hospital, Temerloh 28000, Pahang, Malaysia

**Keywords:** mental health literacy, social stigma, adolescent mental health, mental disorders, Malaysia

## Abstract

Parents, guardians, and teachers are the informal sources of mental health support that adolescents rely on. Nevertheless, limited mental health knowledge limits their ability and confidence in providing appropriate assistance. This study aims to (1) evaluate the relationship between the roles of parents/guardians and teachers and their responses to discover the common misconceptions on mental health among those providing informal support to adolescents and (2) determine which demographic factors would act as the strongest predictor influencing their mental health literacy (MHL) status. The cross-sectional study recruited 867 parents, guardians, and teachers of adolescents from 24 government secondary schools’ parent–teacher associations via multistage stratified random sampling. Parents, guardians, and teachers’ MHL were evaluated using the Mental Health Knowledge Schedule—Malay Version (MAKS-M). The collected data were analyzed using Pearson’s Chi-squared test to investigate the association between the respondents’ roles and responses. Multiple Regression analysis was used to determine the predictors of MHL. The score of MAKS-M for the current study sample is 73.03% (M = 43.82, SD = 4.07). Most respondents responded incorrectly on Items 1 (employment), 6 (help-seeking), 8 (stress), and 12 (grief). Teachers provided more favorable responses on several items than parents and guardians. Finally, younger age, higher income, knowing someone with mental disorders, and having experience of attending formal training on mental health first aid were the significant predictors of MHL. MHL interventions in Malaysia should cater to older adults of lower socioeconomic status and lesser experience in mental health, specifically highlighting the stigmas on mental health help-seeking behaviors, treatment, and employment concerns, plus the recognition of various mental health diagnoses.

## 1. Introduction

According to the Institute for Health Metrics and Evaluation, psychiatric illnesses represent 4.92% of the total disability-adjusted life year, while also being ranked seventh among the causes of global disability-adjusted life year worldwide [1]. Adolescents are deemed to be at risk of developing various mental illnesses due to various possible social factors, such as peer pressure, identity and sexuality conflicts, excessive media exposure, being a victim of violence and abuse, and poor living condition (e.g., poverty, social discrimination, orphaned, and underlying chronic illnesses) [2]. Therefore, the onset of mental illness is usually seen at a young age, such as oppositional-defiant disorder and conduct disorder among younger adolescents and psychosis among their older counterparts [3]. A meta-analysis outlined the peak/median age of onset for all psychiatric disorders as 14.5/18 years, and the proportion for global onset of the first disorder occurred before age 14, 18, and 25 were 34.6%, 48.4%, and 62.5%, respectively [4]. This finding supports the notion that adult psychiatric disorders originate from childhood and peak by their adolescence phase.

However, a study reported that older adolescents tend to rely on themselves to solve their mental struggles and refrained from seeking professional help. Additionally, time and monetary constraints were also the commonly discussed barrier of help-seeking among students [5]. As for their younger counterparts, they were unaware of the available mental health services. Due to the lack of existing interpersonal relationship, students expressed their distrust on the services’ competency and confidentiality. Therefore, they prefer informal sources of mental and emotional support from acquainted individuals, such as peers and family members [6]. Although teachers were rated low in the preferred source of help, their daily interaction with a large group of students may help detect behavioral and emotional changes among teenagers and aid in the proper referral to the available mental healthcare services [6]. Unfortunately, the existing knowledge gap regarding mental health among the adult stakeholders hinders the effort for better communication and provision of assistance for adolescents in psychological distress [7]. Similarly, teachers voiced their concern about the lack of knowledge, formal training, and resources in mental health assistance, causing low confidence in aiding at-risk students [8].

Mental health literacy (MHL) is the comprehension of mental illnesses, which enables the detection, intervention, and prevention of the disorders [9]. Recent reiteration included proper mental health maintenance, adequate comprehension of psychiatric diagnoses and their management, and reduced stigmatization of mental disorders for better help-seeking efficacy [10]. Studies on MHL enable the researchers to objectively establish one’s level of knowledge on mental health care, prevention, and promotion, the knowledge gaps and common stigmas, correlations, and level of effectiveness of MHL interventions [11]. However, the limited epidemiological research on MHL in Malaysia limits our understanding of these issues [12].

Previous MHL research among Malaysian adult stakeholders reported comparable MHL level to the adult population in other countries, such as the general adult population in the United Kingdom, Lebanon, and refugee teachers in Malaysia [13,14,15,16]. However, the MHL levels were found to be higher among caregivers of psychiatric patients in the United Kingdom, Jordanian healthcare professionals, and healthcare providers and volunteers in Kenya [17,18,19]. The disparity was believed to be due to the difference in the level of mental health and care provision exposure. Healthcare providers are formally trained personnel who are required to have in-depth knowledge of the diagnoses and to deliver care without judgment and discrimination [16]. Caregivers, on the other hand, have first-hand experiences in assisting individuals affected by mental disorders and may have more contact with mental health services. The experience can increase their compassion and understanding of mental disorders and strengthen their confidence in mental healthcare services [19].

According to Hurley et al., several notable determinants of help-seeking facilitation include personal cultural and religious backgrounds, financial status, knowledge on mental health, and stigma toward mental health and services [20]. Therefore, the present study aimed to (1) evaluate the relationship between the roles of parents/guardians and teachers of adolescents (in the current study, adolescents referred to those aged between 13 to 17 years old) and their (parents/guardians and teachers) responses to discover the common misconceptions and (2) determine the unique predictors of MHL among West Malaysian parents, guardians, and teachers of adolescents. The study of MHL among this population was expected to discover the common misconceptions and investigate the relevant sociodemographic factors that may aid in the planning and development of gatekeepers’ training on adolescents’ mental health

## 2. Materials and Methods

### 2.1. Study Design and Subjects

The current study is an extraction from a bigger study on mental health literacy. A related manuscript on the association between suicide and mental health literacies has recently been published (see Phoa et al., 2022) [16]. This cross-sectional study involved parent–teacher associations of 24 government secondary schools sampled based on multistage stratified random sampling. The schools were stratified according to the school type (i.e., national secondary schools “SMK”, fully residential schools “SBP”, and religious secondary schools “SMA”) and school locality (i.e., urban and rural settings). Subsequently, the states in West Malaysia were classified into the Northern, Eastern, Central, and Southern zones. For each stratum, one state was randomly sampled from each zone, and one district was similarly sampled from the selected state. Finally, one school was randomly selected from within the chosen district. The minimum sample size of 730 was obtained by using a single proportion formula with 80% statistical power and a significance level of 5%. A design effect of 2 was also applied in the calculation.

The present study included parents, guardians, and teachers of adolescents, Malaysian nationality, and literate in Malay. The respondents were recruited via a research poster linked to the online questionnaire. The posters were initially sent to one key informant from each school, which were then shared among the potential study participants through various social media and messaging applications. In sum, 867 parents, guardians, and teachers of adolescents were recruited from 24 parent–teacher associations of government secondary schools across West Malaysia. The study was conducted between July and September of 2021. The sampling method is summarized and shown in Figure 1.

### 2.2. Instrument

The 12-item Mental Health Knowledge Schedule—Malay Version (MAKS-M), which was back translated from the English version of the Mental Health Knowledge Schedule (MAKS), was utilized to evaluate the level of MHL among the Malay-speaking community [21,22]. The MAKS-M comprised two separate parts. Part A assesses the knowledge of mental health stigma, including stigma on mental health assistance, disorder detection, mental support, employment, management, and recovery. On the other hand, Part B evaluates the respondents’ knowledge of the diagnosis of mental disorders, whereby six different conditions were presented. Participants were required to rate each item on an ordinal scale of 1 to 5 based on their understandings, whereby 1 is “strongly disagree”, and 5 is “strongly agree”. To reflect the accurate answers, Items 6, 8, and 12 were reverse coded. Finally, a total score was calculated out of 60, and a higher MAKS-M score indicated a better MHL level. The original English version of the survey reported moderate internal consistency of 0.65 with moderate to substantial test–retest reliability of 0.71, whereas the MAKS-M reported comparable Cronbach’s alpha at 0.62 [21,22]. The current study yielded an acceptable Cronbach alpha coefficient of 0.68; Part A was 0.54, and Part B was 0.71. The instrument was converted into an online questionnaire using Google Form^®^ (Google, Mountain View, CA, USA) for this study.

### 2.3. Procedure

A multistage stratified random sampling strategy was utilized in sampling the study population. The school administrative board and PIBG committee were contacted to obtain permission to conduct the study. Each school appointed one key informant for convenient communication between the research team and the school. The key informants were requested to assist in disseminating the research recruitment poster and the link to the Google Form among the targeted population. The study information sheet, including the research background, anonymity and confidentiality, potential benefit and harm of the study, voluntary participation, and researchers’ contacts, was attached in the Google Form for the participants’ reference. All completed responses were subsequently collected for further analysis.

### 2.4. Statistical Analysis

The IBM^®^ SPSS^®^ Statistics version 27.0 (IBM, Armonk, NY, USA) was used to analyze the descriptive data of the respondents’ sociodemographic factors. The data were presented in frequency, percentage, mean, and standard deviation. To further clarify the trend of responses and identification of knowledge gaps and misconceptions, the Likert scales from MAKS-M were collapsed into either favorable or non-favorable responses [23]. The association between the respondents’ roles and responses was assessed via Pearson’s Chi-squared analysis for each instrument item. Multiple linear regression analysis was applied to determine the predictors of mental health literacy levels. The variable selection process was carried out through stepwise, forward, and backward methods. The results were presented in regression coefficient (*b*) and 95% CI, and the significance level was chosen at 0.05. The variables’ linear relationship, data distribution, and multicollinearity were also evaluated to meet multiple linear regression analysis assumptions.

### 2.5. Ethical Consideration

The study was undertaken as per the Declaration of Helsinki. Ethics approval from the university’s ethical committee, the Human Research Ethics Committee of Universiti Sains Malaysia, was obtained before the research began (protocol code: USM/JEPeM/21020179). Digital consent for participation and publication was also obtained from each volunteer who wished to participate in this study.

## 3. Results

A total of 882 responses were received by the end of the data collection phase, and only 867 responses were retained after the data clean-up. Fifteen outliers were excluded in view of having a standardized score above 3.29 or below −3.29. The data for each sociodemographic factor were tabulated in Table 1. The respondents’ mean age was 43.81 (SD = 8.34). The majority were parents and guardians (64.6%), female (71.5%), of Malay ethnicity (87.1%), and Muslims (89.9%). As for the socioeconomic status, three-quarters of the respondents received up to a tertiary level education, whereby two-thirds worked in the government sector. The household income stratification based on the Department of Statistics Malaysia showed that 39.9% of respondents were classified under the Bottom 40% (B40) income range, 45.4% earned within the Middle 40% (M40) income range, and only 14.6% of the respondents were categorized within the Top 20% (T20) group. As for their school background information, the current study received more respondents from parent–teacher associations of urban schools (58.1%) than rural schools (41.9%). A total of 39.3% of respondents were from SMK, 37.0% were from SBP, and only 23.6% were from SMA. Finally, the self-reported experience of mental illness reported that only a small number of respondents had experienced mental health issues. Only 2.4% of respondents reported being diagnosed with mental disorders previously, whereas 18.5% had known someone close (e.g., family members, friends, relatives) struggling with mental disorders. One-fourth of the respondents reported being involved in the care or assistance of someone with mental distress. Only 10.1% of the respondents attended formal psychological first aid training.

In general, the current study sample (including parents, guardians, and teachers) scored an overall MAKS-M of 73.03% (M = 43.82, SD = 4.07), whereby parents and guardians scored MAKS-M of 72.75% (M = 43.65, SD = 4.03), while teachers scored slightly higher MAKS-M score at 73.58% (M = 44.15, SD = 4.13). The Pearson’s Chi-squared test analysis was performed to investigate the association between the respondents’ roles and answers. Upon checking, it was revealed that most parents, guardians, and teachers incorrectly answered Item 1 (employment), Item 6 (help-seeking), Item 8 (stress), and Item 12 (grief). Furthermore, Chi-squared analysis found that teachers were significantly associated with more favorable answers as compared to the parents and guardians for several items, i.e., Item 4 (psychotherapy) (*p* = 0.017), Item 6 (help-seeking) (*p* = 0.007), Item 7 (depression) (*p* = 0.004), Item 8 (stress) (*p* = 0.004), Item 9 (schizophrenia) (*p* = 0.002), Item 10 (bipolar) (*p* = 0.001), and Item 12 (grief) (*p* = 0.005). The findings are presented in Table 2.

Simple linear regression was conducted for each sociodemographic factor to determine the variables included in the selection process. The simple linear regression findings are presented in Table 3. Four variables were found to be statistically significant upon stepwise, forward, and backward methods: age, monthly household income bracket, knowing someone with mental disorders, and attended psychological first aid training. The four variables then proceeded with multiple linear regression.

The interaction terms for all the variables were found to be non-significant, thus indicating no significant interactions between all the variables (See Appendix A). Multicollinearity was examined by measuring the variance-inflation factors (VIF). Overall, VIFs of less than 2.5 in this study indicate no multicollinearity between the independent variables, therefore meeting the assumption of multiple linear regression (see Appendix A).

The multiple linear regression analysis is reported in Table 4. According to the final model, it was found that those who had reportedly known someone with mental disorders tend to score a significantly higher mean score of MAKS-M by 1.385; 95% CI (0.691, 2.080). The monthly household income bracket was also a significant predictor of MAKS-M, whereby respondents within the M40 and T20 categories scored higher mean MAKS-M by 0.709; 95% CI (0.122, 1.296) and 1.341; 95% CI (0.502, 2.180), respectively. Additionally, respondents who were 10 years older were found to score significantly lower mean score for MAKS-M by 0.400; 95% CI (−0.073, −0.007). Finally, respondents who reported to have attended formal training were predicted to score a higher mean score of MAKS-M by 1.016; 95% CI (0.134, 1.904).

Final model:MAKS-M = 44.025 + 1.385 (known someone with mental disorder) + 1.341 (income T20) + 1.016 (attended formal training) + 0.709 (income M40) − 0.040 (age)

## 4. Discussion

### 4.1. Common Stigma and Misconceptions

Upon dichotomizing the responses, it was observed that most respondents had low knowledge of stigma about employment and help-seeking behaviors among psychiatric patients. Psychiatric patients were more likely to be discriminated against from work integration, leading to a higher rate of unemployment, lower job security, and limited career growth opportunities compared to everyone else. It was thought to be due to the belief that individuals affected by mental illnesses tended to be violent, incompetent, and had poor coping abilities [24]. In Malaysia, the lack of employment prospects for those with mental disorders was reported to affect their confidence, thus, leaving them susceptible to poverty, worsening their self-stigma, and compromising their treatment progression [25].

Subsequently, the stigmatization of mental health help-seeking behaviors was also reported in previous studies. People did not wish to be labeled as “weak” and were embarrassed by society’s perception of receiving mental healthcare services [6]. Additionally, Razali and Najib reported that two-thirds of Malay psychiatric patients prefer to consult traditional healers before receiving mental health evaluations from healthcare professionals [26]. It is likely due to the ethnocultural and religious beliefs that mental disorders were caused by divine intervention, black magic, or possession by evil spirits [27]. The stigma against mental health services and misconceptions about psychiatric disorders should be rectified to reduce social discrimination and encourage appropriate mental health help-seeking behaviors.

For Part B of the instrument, very few respondents correctly identified stress (MAKS-M 8) and grief (MAKS-M 12) as non-psychiatric diagnoses. It was expected that stress and grief to be falsely categorized as mental illnesses due to their strong relationship with psychiatric illnesses. For instance, a strong association was observed between daily stressors and adverse mental health, symptoms of depression, anxiety, and stress [28]. Similarly, a prolonged state of acute grief (i.e., complicated grief) affects approximately 10% of bereaved individuals. It could lead to an increased risk of physical complications (e.g., cardiac problems, hypertension, cancer), psychiatric disorders (e.g., addiction, drug abuse, depression), and increased risk of suicidal behaviors [29]. Although both stress and grief were described as emotional and mental responses to adverse circumstances and loss, neither was considered a mental health diagnosis.

### 4.2. Association between Participants’ Roles and their Responses

This study also found that parents and guardians were associated with less favorable responses for Items 4, 6, 7, 8, 9, 10, and 12. Parents and guardians have a higher pre-existing stigma on psychotherapies and mental health help-seeking behaviors. They also could not identify depression, schizophrenia, and bipolar disorder as psychiatric diagnoses and falsely labeled stress and grief as mental illnesses. This finding can be attributed to the difference in the level of education among parents or guardians versus educators. In studies among people with diabetes, health literacy was discovered to be a significant mediator between education level and health outcomes [30,31]. Similarly, higher education level was reportedly linked to better MHL in terms of less stigma and improved detection of psychiatric disorders [32]. In Malaysia, teachers are required to undergo a minimum of undergraduate-level teaching degrees. Therefore, almost all the teachers in the current sample received up to a tertiary education. Furthermore, teachers’ exposure to basic knowledge of child and adolescents’ mental health, learning disabilities, and educational psychology throughout their undergraduate training and teaching experiences may enhance their awareness and understanding of the common mental illnesses among adolescents [33,34]. Nonetheless, most educators voiced their concerns about their limited training and knowledge and are eager to improve themselves. Various studies underscored the need for more extensive mental health first aid training among educators to improve their understanding and confidence in providing mental health assistance [35,36,37].

### 4.3. Predictors of MHL

Regression analysis revealed that respondents’ age was one of the critical predictors of MHL level, with younger respondents having greater MHL levels. This result is consistent with findings from earlier research among Jordanian and Lebanese populations [15,17]. Similarly, a systematic review reported that elderly Malaysians consider mental health services to be ineffective compared to the younger generation. Therefore, they tend to be more reluctant to seek psychological help [38]. The development of mental health care policies and services through time may cause this phenomenon. The Mental Health Act 2001 (Act 615) was put into effect in Malaysia, whereby mental healthcare transitioned from institutional treatment to more therapeutic and community-based care. The implementation of the National Mental Health Policy in 1998 and the Mental Health Framework in 2001 also made it possible for Malaysians to obtain more wide-ranging mental healthcare treatments. In addition to community-based mental health treatments for rehabilitating and reintegrating stable patients back into society, the Malaysian mental healthcare system is currently said to have good integration throughout primary, secondary, and tertiary care [39]. These transformations might improve knowledge and lessen social stigma among the younger generation. Thus, MHL intervention designs should cater to the needs of the older generation to improve their knowledge and assurance toward mental health services.

Our study also demonstrated that individuals with higher monthly household incomes have a higher knowledge of mental health diagnoses and general level of MHL. We discovered that those in the B40 income category performed significantly poorer than their counterparts. Our findings lend support to previous studies, which found that having a higher socioeconomic position was an important determinant of improved mental health literacy [19,40]. Likewise, Munawar and colleagues reported that the relationship between economic factors and psychiatric issues in Asian countries was well documented [38]. These outcomes might reflect the disparity between the number of mental health services and awareness programs available for various socioeconomic statuses. Therefore, accessibility and affordability should be considered when planning MHL intervention and awareness activities.

Higher MHL levels were predicted among those with experience of mental disorders and interventions (i.e., those who have known someone who has a psychiatric disorder and has taken formal psychological first aid training). This finding was in line with a study by Razali and Ismail, whereby having close family or friends with mental disorders showed a significant reduction in negative attitudes, social discrimination, and improved toleration toward psychiatric patients [41]. Furthermore, several studies observed that formal mental health interventions effectively improved MHL and demystified mental health issues [42,43]. The experience of providing care was believed to strengthen empathy and compassion. As a result, the stigma associated with mental illness was better understood, and prejudice towards those who suffer from mental diseases declined.

### 4.4. Strengths and Limitations

To the best of our knowledge, our study is the first to explore the level of MHL among parents and teachers of adolescents. A systematic review on MHL research in Malaysia reported studies among the general Malaysian community, caregivers of mentally ill patients, healthcare providers, and school or university students [38]. Additionally, multistage stratified random sampling allows a thorough and non-biased sampling across West Malaysia. However, the current study is limited by the cross-sectional study design, whereby causal inferences cannot be made according to the findings if the exposure was developed over time [44]. Furthermore, there is a lack of responses from the Malaysian racial minorities, which may influence the generalizability of the findings. Our study best represents the Malay Muslim community, the majority of the West Malaysian population. The inclusion of vernacular schools is recommended for future studies. Involvement of the East Malaysian population could also improve our understanding of the difference in the ethnocultural views on mental health among the indigenous communities. Finally, although the MAKS-M was psychometrically valid and culturally adapted to the local community, the instrument did not assess all sub-components of MHL as defined previously. Instruments evaluating the MHL level should be improved to incorporate items that assess knowledge, beliefs, help-seeking attitudes, and stigma relating to common psychiatric disorders [38].

## 5. Conclusions

In summary, parents, guardians, and teachers have a relatively similar MHL level. However, parents and guardians responded less favorably to several items in the instrument, indicating possible existing stigma on psychotherapies and help-seeking behaviors among psychiatric patients. Additionally, teachers were better able to correctly label depression, schizophrenia, and bipolar disorder as mental disorders than the parents and guardians. Parents and guardians also tend to misidentify stress and grief as mental disorders as opposed to the educators. Finally, older age, lower socioeconomic status, and lesser experience in mental health predict a lower level of MHL among the targeted population. These findings should aid policymakers and program management to form an evidence-based intervention program framework and policies to tackle specific facets of mental health knowledge gaps and target those who are predicted to have a lower understanding of mental health. Further investigation via a qualitative approach is commended for understanding the subjective opinions of parents, guardians, and teachers on the barriers and facilitators of mental health knowledge acquisition.

## Figures and Tables

**Figure 1 ijerph-20-00825-f001:**
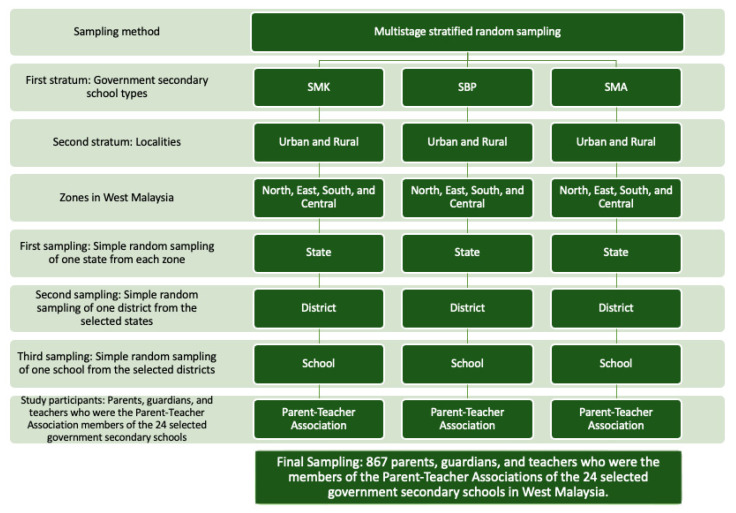
Multistage stratified random sampling for the current study.

**Table 1 ijerph-20-00825-t001:** Descriptive analysis of each sociodemographic factor of the study population (n = 867).

Variables	n (%)	Mean (SD)
*Age* (years)		43.81 (8.34)
*Role*		
Parents or caretakers	560 (64.6)	
Teacher	307 (35.4)	
*Sex*		
Male	247 (28.5)	
Female	620 (71.5)	
*Ethnicity*		
Malay	755 (87.1)	
Chinese	49 (5.7)	
Indian	4 (0.5)	
Other Bumiputera	48 (5.5)	
Others	11 (1.3)	
*Religion*		
Islam	779 (89.9)	
Christian	49 (5.7)	
Hindu	3 (0.3)	
Buddhist	34 (3.9)	
Others	2 (0.2)	
*Education level*		
No formal education	2 (0.2)	
Primary education	11 (1.3)	
Secondary education	202 (23.3)	
Tertiary education	652 (75.2)	
*Occupation sector*		
Unemployed/homemaker	102 (11.8)	
Government	578 (66.7)	
Private	106 (12.2)	
Self-employed	63 (7.3)	
Pensioner	18 (2.1)	
*Monthly household income bracket ^a^*		
B40	346 (39.9)	
M40	394 (45.4)	
T20	127 (14.6)	
*School locality*		
Urban	502 (58.1)	
Rural	363 (41.9)	
*School type*		
SMK	341 (39.3)	
SBP	321 (37.0)	
SMA	205 (23.6)	
*Personal history of mental illness*		
Yes	21 (2.4)	
No	846 (97.6)	
*Had known someone with mental illness*		
Yes	160 (18.5)	
No	707 (81.5)	
*Had assisted those with mental illness*		
Yes	217 (25.0)	
No	650 (75.0)	
*Had attended formal psychological first aid training*		
Yes	88 (10.1)	
No	779 (89.9)	

Note: Adapted from “The Malay Literacy of Suicide Scale: A Rasch Model Validation and Its Correlation with Mental Health Literacy among Malaysian Parents, Caregivers and Teachers”, by Phoa, PKA et al., 2022, Healthcare, Vol. 10, No. 7, p. 1304). n = frequency, SD = standard deviation, SMK = national daily secondary schools, SBP = fully residential schools, SMA = religious secondary schools. ^a^ Malaysian household income stratification: Bottom 40% (B40) (below RM4850); Middle 40% (M40) (RM4950–RM10959); Top 20% (T20) (RM10960 and above).

**Table 2 ijerph-20-00825-t002:** Association between respondents’ roles and their responses based on favorable and non-favorable answers (n = 867).

Role	Item	Dimension	Part	Favorable Responses n (%)	Non-Favorable Responses n (%)	Mean (SD)
Parents and guardians (n = 560)	MAKS-M 1	Employment	A (Knowledge of mental health stigma)	262 (46.8)	298 (53.2)	43.65 (4.03)
	MAKS-M 2	Support		466 (83.2)	94 (16.8)	
	MAKS-M 3	Psychotropic drugs		291 (52.0)	269 (48.0)	
	MAKS-M 4 *	Psychotherapy		475 (84.8)	85 (15.2)	
	MAKS-M 5	Recovery		298 (53.2)	262 (48.0)	
	MAKS-M 6 **	Help-seeking		91 (16.3)	469 (83.7)	
	MAKS-M 7 **	Depression	B (Knowledge on the diagnosis of mental disorders)	508 (90.7)	52 (9.3)	
	MAKS-M 8 **	Stress		24 (4.3)	536 (95.7)	
	MAKS-M 9 **	Schizophrenia		442 (78.9)	118 (21.1)	
	MAKS-M 10 **	Bipolar		444 (79.3)	116 (20.7)	
	MAKS-M 11	Addiction		369 (65.9)	191 (34.1)	
	MAKS-M 12 **	Grief		46 (8.2)	514 (91.8)	
Teachers (n = 307)	MAKS-M 1	Employment	A (Knowledge of mental health stigma)	146 (47.6)	161 (52.4)	44.15 (4.13)
	MAKS-M 2	Support		245 (80.0)	62 (20.0)	
	MAKS-M 3	Psychotropic drugs		171 (55.7)	136 (44.3)	
	MAKS-M 4 *	Psychotherapy		278 (90.6)	29 (9.4)	
	MAKS-M 5	Recovery		176 (57.3)	131 (42.7)	
	MAKS-M 6 **	Help-seeking		73 (23.8)	234 (76.2)	
	MAKS-M 7 **	Depression	B (Knowledge on the diagnosis of mental disorders)	295 (96.1)	12 (3.9)	
	MAKS-M 8 **	Stress		28 (9.1)	279 (90.9)	
	MAKS-M 9 **	Schizophrenia		268 (87.3)	39 (12.7)	
	MAKS-M 10 **	Bipolar		272 (88.6)	35 (11.4)	
	MAKS-M 11	Addiction		196 (63.8)	111 (36.1)	
	MAKS-M 12 **	Grief		44 (14.3)	263 (86.7)	
	The overall mean score of the study population					43.82 (4.07)

Note: n = frequency, MAKS-M = Mental Health Knowledge Schedule—Malay Version, SD = standard deviation. * Significant at *p*-value ≤ 0.05 level, ** significant at *p* ≤ 0.01 level.

**Table 3 ijerph-20-00825-t003:** Simple linear regression of mean score of MAKS-M among parents, guardians, and teachers of adolescents in West Malaysia (n = 867).

Variables	Simple Linear Regression		
	*b*	95% CI	*p*-Value
Known someone with mental health disorder ^a, b^	1.588	0.896, 2.280	<0.001
Monthly household income bracket ^a, b^	0.602	0.213, 0.991	0.002
Age ^a, b^	−0.032	−0.064, 0.001	0.057
Attended formal psychological first aid training ^a, b^	1.221	0.325, 2.116	0.008
Role ^a^	−0.500	−1.067, 0.067	0.084
Sex ^a^	−0.682	−1.282, −0.082	0.026
Ethnicity ^a^	−0.271	−0.596, 0.054	0.102
Religion ^a^	−0.463	−0.997, 0.071	0.089
Education level ^a^	0.682	0.123, 1.241	0.017
Occupation sector	−0.032	−0.363, 0.299	0.851
School locality ^a^	0.403	−0.147, 0.952	0.151
School type ^a^	−0.137	−0.486, 0.212	0.440
Personal diagnosis of mental health disorders ^a^	1.157	−0.608, 2.921	0.199
Assisted someone with mental health disorders ^a^	0.746	0.121, 1.370	0.019

Note: MAKS-M = Mental Health Knowledge Schedule—Malay Version, 95% CI = 95% confidence interval, *b* = crude regression coefficient, Adj. *b* = adjusted regression coefficient. ^a^ Variables included in the stepwise linear regression for the variable selection process. ^b^ Significant variables selected for multiple linear regression upon the variable selection process.

**Table 4 ijerph-20-00825-t004:** Multiple linear regression of mean score of MAKS-M among parents, guardians, and teachers of adolescents in West Malaysia (n = 867).

Variables	Multiple Linear Regression		
	Adj. *b*	95% CI	*p*-Value
Known someone with a mental health disorder (1 = Yes, 0 = No)	1.385	0.691, 2.080	<0.001
Monthly household income bracket			
B40			
M40	0.709	0.122, 1.296	0.018
T20	1.341	0.502, 2.180	0.002
Age	−0.040	−0.073, −0.007	0.018
Attended formal psychological first aid training (1 = Yes, 0 = No)	1.016	0.134, 1.904	0.025

Note: MAKS-M = Mental Health Knowledge Schedule—Malay Version, 95% CI = 95% confidence interval, Adj. *b* = adjusted regression coefficient, B40 = Bottom 40%, M40 = Middle 40%, T20 = Top 20%.

## Data Availability

Not applicable.

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
