# Peer review of "Predictors of Mental Health Literacy among Parents, Guardians, and Teachers of Adolescents in West Malaysia"

_ijerph, 2023, doi:10.3390/ijerph20010825_

Round 1

Reviewer 1 Report (Previous Reviewer 1)

Thank you for taking into account my suggestions.

However, you have not completely responded to my first suggestion. When I wrote "dimension to which these points refer to" I was talking about the items 4, 6, 7, 8, 9, 10, 12 (Lines 212-213 in the new manuscript).

Please, consider adding them.

Author Response

Reviewer 2 Report (Previous Reviewer 2)

The authors addressed my comments satisfactorily, with the exception of the following points:  
  1. For my comment 6, the authors stated that "The 15 responses were considered as outliers, and therefore were excluded in the study." Outliers should NEVER be excluded from the study. If they are verified as correct answers, then the analysis should be presented with and then without the outliers, and the findings contrasted and compared. Most likely, a non-parametric analysis will have to be performed with the outliers included in the dataset.
  2. In Table S2, the VIF for age is in the thousands, and for training is missing. You need to check those numbers again.
  3. Why did you choose the cutoff point of 10 for the VIF when it should be 2.5? Please see for reference doi:10.1007/s11135-017-0584-6
  4. Effect modifiers should be categorical variables tested for the purpose of performing subgroup analyses in the case they present significant interaction with the potential exposure variable. No interaction between independent variables/confounders should be performed. Testing for interaction should be done one step at the time, and not multiple interactions tested simultaneously. A clear description of the methods used to test for effect modification should be included in the paper, and the results discussed.

Author Response

Reviewer 3 Report (Previous Reviewer 3)

I have no more comments. I endorse this for publication.

Author Response

Thank you, Reviewer 3 for your review.

Round 2

Reviewer 2 Report (Previous Reviewer 2)

I have no more comments to make.

This manuscript is a resubmission of an earlier submission. The following is a list of the peer review reports and author responses from that submission.

Round 1

Reviewer 1 Report

The article "Predictors of Mental Health Literacy among Parents, Guardians, and Teachers of School-going Adolescents in West Malaysia", aims to evaluate the sociodemographic factors that could predict the Mental Health Literacy among West Malaysian parents, caregivers, and teachers of school-going adolescents.

The manuscript is well written and has been pleasant to read. However, I have some comments listed below:

1-   Lines 196,198-199: Write the dimension to which these points refer to improve the clarity of the text in the following sections.

2-   Lines 219, 222, 224, 226: Put the CI values between brackets. E.g. 1.385; 95% CI (0.691, 2.080).

3-   Lines 228-229: I recommend listing them from highest to lowest OR value for clarity.

Reviewer 2 Report

This is an interesting and needed paper given the lack of literature in this area for West Malaysia. That being said, I do have multiple concerns about this paper:

1. On line 104 you got me confused. Please explain in great details what your population under investigation is. To this end, please be very specific regarding the person, place and time aspects.

2. On lines 109-110 the authors wrote "A dropout rate of 20% and a design effect of 2 were also applied in the calculation." while on line 100 the authors wrote "This cross-sectional study ...". Please explain why you applied a dropout rate for a cross-sectional study. This is not clear.

3. On line 115 the authors wrote "In sum, 24 schools and 867 parents and teachers from the ...". So, based on your sentence, there were no caregivers enrolled in your study? A clear differentiation needs to be made between parents and caregivers and the specific numbers need to be presented.

4. You need to include the corresponding reference on line 119 for the Malay Version MAKS-M.

5. On line 127 the authors wrote "Items 6, 8, and 12 were reverse coded". Why is that the case? A reverse coding may confuse the study participants leading them to give the wrong answers.

6. On lines 165-166 the authors wrote "... and 165 only 867 responses were retained after the data clean-up". The exclusion criteria need to be clearly specified, with the corresponding number of individuals excluded for each of the selected criteria.

7. The sentence from lines 191 through 193 is not clear. The current study population or the current study sample? They scored what? On what? Not clear at all.

8. On lines 195-196 the authors wrote "Upon checking, it was revealed that most parents, caregivers, and teachers incorrectly answered Item 1, Item 6, Item 8, and Item 12". Given my comment 5, this doesn't seem to surprise me. How did you solve this problem?

9. On lines 196-199 the authors present chi-squared analysis results. Are these based on the incorrectly answered questions for items 6, 8 and 12. How can that be?

10. On lines 228-229 your population model is not complete. Where is your error term. And have you checked your residual assumptions for the validity of your linear regression model? Also, have you tested for the effect modifiers of your model?

Reviewer 3 Report

attached.
